# The impact of lag time to cancer diagnosis and treatment on clinical outcomes prior to the COVID-19 pandemic: A scoping review of systematic reviews and meta-analyses

Parker Tope[1], Eliya Farah[1], Rami Ali[1], Mariam El-Zein[1], Wilson H Miller[2], Eduardo L Franco[1]*

[1]Division of Cancer Epidemiology, McGill University, Montreal, Canada; [2]Department of Oncology, McGill University, Montreal, Canada

*For correspondence:
eduardo.franco@mcgill.ca

## Abstract

**Background:** The COVID-19 pandemic has disrupted cancer care, raising concerns regarding the impact of wait time, or 'lag time', on clinical outcomes. We aimed to contextualize pandemic-related lag times by mapping pre-pandemic evidence from systematic reviews and/or meta-analyses on the association between lag time to cancer diagnosis and treatment with mortality- and morbidity-related outcomes.

**Methods:** We systematically searched MEDLINE, EMBASE, Web of Science, and Cochrane Library of Systematic Reviews for reviews published prior to the pandemic (1 January 2010–31 December 2019). We extracted data on methodological characteristics, lag time interval start and endpoints, qualitative findings from systematic reviews, and pooled risk estimates of mortality- (i.e., overall survival) and morbidity- (i.e., local regional control) related outcomes from meta-analyses. We categorized lag times according to milestones across the cancer care continuum and summarized outcomes by cancer site and lag time interval.

**Results:** We identified 9032 records through database searches, of which 29 were eligible. We classified 33 unique types of lag time intervals across 10 cancer sites, of which breast, colorectal, head and neck, and ovarian cancers were investigated most. Two systematic reviews investigating lag time to diagnosis reported different findings regarding survival outcomes among paediatric patients with Ewing's sarcomas or central nervous system tumours. Comparable risk estimates of mortality were found for lag time intervals from surgery to adjuvant chemotherapy for breast, colorectal, and ovarian cancers. Risk estimates of pathologic complete response indicated an optimal time window of 7–8 weeks for neoadjuvant chemotherapy completion prior to surgery for rectal cancers. In comparing methods across meta-analyses on the same cancer sites, lag times, and outcomes, we identified critical variations in lag time research design.

**Conclusions:** Our review highlighted measured associations between lag time and cancer-related outcomes and identified the need for a standardized methodological approach in areas such as lag time definitions and accounting for the waiting-time paradox. Prioritization of lag time research is integral for revised cancer care guidelines under pandemic contingency and assessing the pandemic's long-term effect on patients with cancer.

**Funding:** The present work was supported by the Canadian Institutes of Health Research (CIHR-COVID-19 Rapid Research Funding opportunity, VR5-172666 grant to Eduardo L. Franco). Parker

Tope, Eliya Farah, and Rami Ali each received an MSc. stipend from the Gerald Bronfman Department of Oncology, McGill University.

## Editor's evaluation

The scoping review has been constructed with novel time references for specific cancer treatment progressions. The extent of novel thought aggregating the literature makes an outstanding scientific contribution.

## Introduction

The sudden toll of the coronavirus disease of 2019 (COVID-19) pandemic on healthcare systems worldwide has transformed the provision of cancer control and care services. With successive waves of SARS-CoV-2 variants waxing and waning disparately between and within countries, the standard cancer care framework from diagnosis to treatment has been distorted. Resulting stressors on cancer centres have introduced a multitude of challenges, including prolongation of existing lag times within the cancer care continuum. In the early months of the pandemic, cancer screening services were temporarily suspended (*Mast and Munoz del, 2020*; *Maringe et al., 2020*; *Villain et al., 2021*). By May 2020, two months after the World Health Organization declared the COVID-19 outbreak a pandemic, the volume of colorectal and breast cancer screenings in the United States dropped by 85% and 94%, respectively, compared to averages from the previous three years (*Mast and Munoz del, 2020*). Routine and diagnostic patient visits to primary healthcare providers have been similarly affected. Clinics' reduced overall patient volume and patient hesitancy to seek in-person care have further contributed to the backlog of elective and non-elective cancer services and procedures (*Walker et al., 2022*; *Lesley et al., 2021*; *Jazieh et al., 2020*). New and previously diagnosed cancer cases have been subject to risk-based prioritization and treatment triaging that differ from standard clinical practice (*Farah et al., 2021*). For cancers that were diagnosed in the first six months of the pandemic, an increasing trend in diagnosis of late-stage cancers and a decreasing trend in diagnosis of early-stage cancers were observed, as a reflection of the initial effects of extended lag times to diagnosis (*Carvalho et al., 2022*).

Changes in dosing and fractionation, as well as delays and interruptions in chemotherapy and radiotherapy regimens of palliative and curative intent care have altered the sensitivity and timeliness of treatment administration (*Teckie et al., 2021*; *Elkrief et al., 2020*). In addition, COVID-19 safety measures within hospitals and cancer centres have congested surgical windows, leaving surgical treatment opportunities for only the most urgent, non-elective cases (*Moletta et al., 2020*). Delay or cessation of clinical trials have restricted treatment and research opportunities integral for both present and future patients with cancer (*Sharpless, 2020*). Cumulatively, pandemic-dictated modifications to standard protocols for radiologic, surgical, and systemic therapy interventions have exacerbated lag times to cancer treatment.

Pandemic-induced changes have brought forth the rising concern within the cancer care community as to whether current lag times to diagnosis and treatment that deviate from standard-of-care practice will lead to poorer outcomes for cancer patients. With modelling studies forecasting the tolerability of these lag times based on estimated long-term outcomes (*Sud et al., 2020*; *Burger et al., 2021*; *Malagón et al., 2022*) and a recent scoping review summarizing the impact of the pandemic on time to cancer diagnosis and treatment (*Carvalho et al., 2022*), a pre-pandemic perspective of the effect of lag time on oncologic outcomes among patients undergoing cancer screening, diagnosis, and staging is of paramount importance. Retrospective, pre-pandemic data can help our understanding of the influence of lag time on patient outcomes, which would be imperative for planning cancer control and care services in the future. Thus, the purpose of this scoping review is to contextualize pandemic-related lag times to cancer diagnosis and treatment by presenting an overview of aggregated pre-pandemic data from systematic reviews and/or meta-analyses on the association between lag time to cancer diagnosis and treatment and clinical outcomes.

## Methods

Results from this review were reported in accordance with the Preferred Reporting Items for Systematic Reviews and Meta-analyses extension for Scoping Reviews (PRISMA-ScR) guidelines (*Tricco et al., 2018*).

### Search strategy and selection criteria

We systematically searched four electronic databases: MEDLINE, EMBASE, Web of Science, and the Cochrane Library of Systematic Reviews. The search strategy consisted of the index keywords *cancer*, *diagnosis* & *treatment*, *wait-time*, *delay*, *outcome*, and *systematic review* and/or *meta-analysis*, along with their associated MeSH and iterative search terms (*Supplementary file 1*). We queried these databases for records published between 1 January 2010 and 31 December 2019. We chose the former date to avoid capturing changes in treatment modalities for cancer sites that might have affected clinical outcomes, and the latter date to prevent artifacts from the COVID-19 pandemic era impacting our search findings. We did not apply language restrictions. In addition, we manually searched the reference lists of eligible systematic reviews and/or meta-analyses to identify potentially relevant reviews missed in our search.

### Eligibility assessment

We imported all records into EndNote X9 reference management software where duplicates were removed. Subsequently, remaining records were uploaded to Rayyan web tool for systematic reviews (*Ouzzani et al., 2016*), where additional duplicates were removed. To be included, a review needed to (1) be a systematic review and/or meta-analysis, (2) refer to any clinical outcome associated with a lag time to cancer diagnosis or treatment, and (3) mention lag time to cancer diagnosis or treatment. In the first round, two reviewers (PT and EF; PT and RA; or EF and RA) independently screened the records' titles and abstracts for eligibility in Rayyan. Discrepancies were resolved through discussion. Full-text eligibility screening was performed independently by two reviewers (EF and RA) and validated by a third (PT). Reference lists of records included in the full-text eligibility screening were manually searched by two reviewers (EF and RA) and validated by a third (PT) for further records that met the eligibility criteria.

### Data abstraction

The included records were divided into two sets and data were independently abstracted by two reviewers (PT and EF for one set, and PT and RA for the second set). One reviewer (PT) then verified all abstracted data. Inconsistencies among reviewers were resolved through discussion. From all included literature, we extracted the following variables: databases searched, number of hits, number of included studies, total number of participants, countries in which studies were conducted, start and endpoints of lag times evaluated, range of the lag time interval, cancer site, cancer type, and outcomes (e.g., mortality, disease-free survival [DFS], recurrence-free survival [RFS], disease progression, etc.). We extracted additional information and variables particular to the review type. This included overall qualitative findings from systematic reviews without a meta-analytic component (referred to herein as systematic reviews). For reviews with a meta-analytic component (referred to hereafter as meta-analyses), we further extracted lag time variable type (categorical or continuous), lag time comparator and reference categories, corresponding pooled risk estimates (e.g., risk ratios [RRs], hazard ratios [HRs], and odds ratios [ORs]) with their respective 95% confidence intervals (CIs), model parameters, heterogeneity statistics, and results from subgroup (by cancer type, study quality, follow-up period, and confounding adjustment, i.e., for sex, smoking, alcohol consumption, study design, and sociodemographic variables) and/or sensitivity analyses. Findings from meta-analyses were categorized by outcome: morbidity- (e.g., disease progression) or mortality- related.

### Definition and representation of lag time intervals

Based on the extracted start and endpoints of lag times, we defined unique lag time intervals that encompassed diagnostic, system, and treatment wait times. We visually represented these lag time intervals on a timeline by including horizontal bars that align the start and endpoints in relation to relevant 'milestones' along the cancer care continuum (i.e., symptom onset, first seen by primary care physician, referral to a specialist, diagnosis, treatment, and palliative care). For reviews that referred

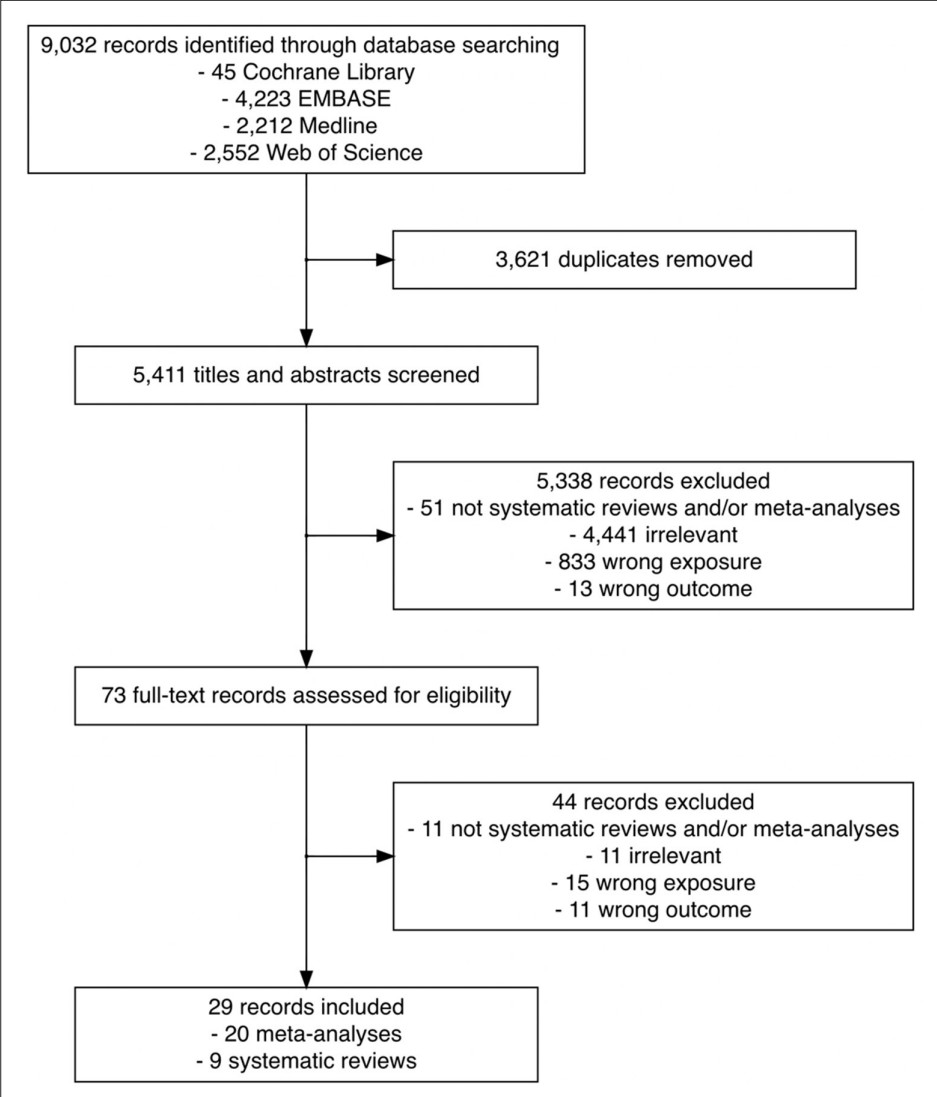

**Figure 1.** Flowchart of the search and study selection of systematic reviews and meta-analyses on the association between time to cancer diagnosis and/or treatment and clinical outcomes.

to a lag time of interest using terms of common usage in the cancer community (e.g., diagnostic delay and treatment delay), we abstracted the start and endpoints of lag time intervals from the original studies and cross-validated these definitions with those used in the included review (*Seoane et al., 2016*; *Seoane et al., 2012*; *Neal et al., 2015*). Some reviews did not mention whether the time between the start and endpoints included or excluded other 'milestones' along the cancer care continuum (*Neal et al., 2015*; *Zhao et al., 2019*; *Hangaard Hansen et al., 2018*; *Graboyes et al., 2019*; *van den Bergh et al., 2013*). We used orange shading to differentiate these from lag time intervals that included milestones in between start and endpoints which are shaded in blue (*Seoane et al., 2016*; *Seoane et al., 2012*; *Neal et al., 2015*; *Graboyes et al., 2019*; *Warren et al., 2019*; *Foster et al., 2013*; *Mattosinho et al., 2019*; *Brasme et al., 2012*; *Lethaby et al., 2013*; *Doubeni et al., 2018*; *Gupta et al., 2016*; *Wu et al., 2018*; *Wang et al., 2016*; *Petrelli et al., 2016*; *Petrelli et al., 2019*; *Du et al., 2018*; *Gómez et al., 2009*; *Lin et al., 2016*; *Loureiro et al., 2016*; *Yu et al., 2013*; *Raphael et al., 2016*; *Zhan et al., 2018*; *Des Guetz et al., 2010*; *Biagi et al., 2011*; *Usón et al., 2017*).

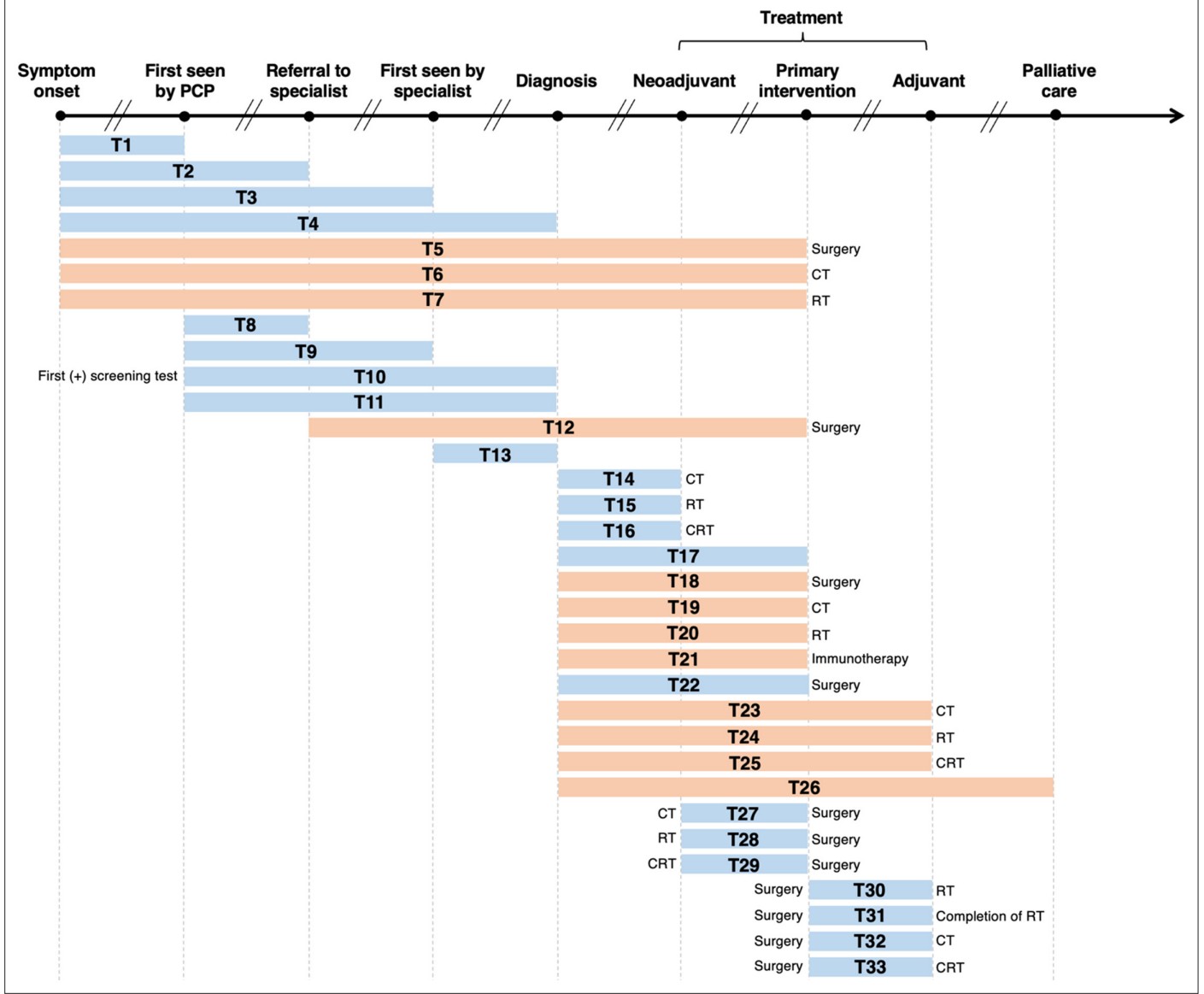

**Figure 2.** Visualization of lag time intervals identified in systematic reviews and/or meta-analyses on the association between time to cancer diagnosis and/or treatment and clinical outcomes. Top arrow represents the cancer care continuum along broad milestones (data points in bold). Oblique breaks denote the incongruency of lag times between milestones (i.e., inconsistent periods of time between milestones; not every cancer patient undergoes all milestones or undergo each milestone sequentially). Each bar indicates a lag time interval (T1–T33). Start and endpoints of each lag time interval are defined by the corresponding milestones. Text before or after a bar defines specific start or endpoints of a lag time interval whenever explicitly reported in a systematic review and/or meta-analysis. Orange shading of bars denotes lag time intervals that do not necessarily include all the milestones through which the bars physically pass (e.g., T18 starts at diagnosis and ends at surgery, without necessarily including neoadjuvant therapy). Blue shading of bars denotes lag time intervals that do include all milestones through which the bars physically pass (e.g., T22 starts at diagnosis and ends at surgery, including neoadjuvant therapy). ART, adjuvant radiotherapy; CRT, chemoradiotherapy; CT, chemotherapy; PCP, primary care provider; RT, radiotherapy.

## Results

*Figure 1* outlines the search results for relevant literature and review selection. Of 9032 records identified, 3621 duplicates were removed, leaving 5411 records for primary screening. Based on title and abstract screening, we excluded 5338 records that did not meet the eligibility criteria. We retrieved the full text of 73 records and further excluded 44. We did not identify any eligible records via manual search of the reference lists of the 73 eligible records. A total of 20 meta-analyses and 9 systematic reviews were included in the current scoping review.

**Table 1.** Lag time intervals evaluated in systematic reviews and meta-analyses on the association between time to diagnosis and/or treatment and clinical outcomes, by cancer site.

| Cancer site | Lag time interval* | First author (year), type of review |
|---|---|---|
| | T30 | *Loureiro et al., 2016*, meta-analysis |
| Brain | T33 | *Warren et al., 2019*, systematic review |
| | T32 | *Yu et al., 2013*, meta-analysis |
| | T30 | *Gupta et al., 2016*, meta-analysis |
| | T32 | *Raphael et al., 2016*, meta-analysis |
| Breast | T32 | *Zhan et al., 2018*, meta-analysis |
| Blood | T21 | *Zhao et al., 2019*, meta-analysis |
| | T32 | *Des Guetz et al., 2010*, meta-analysis |
| | T32 | *Biagi et al., 2011*, meta-analysis |
| | T29 | *Foster et al., 2013*, systematic review |
| | T29 | *Wang et al., 2016*, meta-analysis |
| | T29 | *Petrelli et al., 2016*, meta-analysis |
| | T29 | *Du et al., 2018*, meta-analysis |
| | T28 | *Wu et al., 2018*, meta-analysis |
| | T18 | *Hangaard Hansen et al., 2018*, meta-analysis |
| Colorectal | T32 | *Petrelli et al., 2019*, meta-analysis |
| Eye | T4 | *Mattosinho et al., 2019* meta-analysis |
| | T4 | *Gómez et al., 2009*, meta-analysis |
| | T1, T2, T4, T11 | *Seoane et al., 2012*, meta-analysis |
| | T1, T2, T4, T11 | *Seoane et al., 2016*, meta-analysis |
| | T29 | *Lin et al., 2016*, meta-analysis |
| Head and neck | T17, T30, T31 | *Graboyes et al., 2019*, systematic review |
| | T4 | *Brasme et al., 2012*, systematic review |
| Paediatric | T4 | *Lethaby et al., 2013*, systematic review |
| Prostate | T18, T20 | *van den Bergh et al., 2013*, systematic review |
| | T32 | *Liu et al., 2017*, meta-analysis |
| Ovarian | T32 | *Usón et al., 2017*, meta-analysis |
| | T1–T9, T11–T16, T18–T27 | *Neal et al., 2015*, systematic review |
| Multisite | T10 | *Doubeni et al., 2018*, systematic review |

*Lag time intervals correspond to those defined in **Figure 2**. Reviews on the same cancer site are sorted by publication year.

As illustrated in **Figure 2**, we labelled 33 unique lag time intervals. A plurality of the identified lag time intervals encompassed lag time experienced during treatment. Notably, few reviews specified whether common milestones in care were experienced between defined start and endpoints. For example, although T18 and T22 had similar start and endpoints (from diagnosis to primary intervention by surgery, respectively), T18 is depicted in orange as it was specifically stated that studies were excluded from the review if patients underwent neoadjuvant therapy after diagnosis and before surgery (**van den Bergh et al., 2013**), while T22 is depicted in blue as the corresponding review included studies of patients who underwent neoadjuvant therapy prior to surgery (**Neal et al., 2015**).

**Table 1** lists, by cancer site, the lag time intervals shown in **Figure 2**, where a clear pattern can be observed. Most meta-analyses on breast cancer examined the time between surgery and adjuvant

chemotherapy (ACT; T32) (*Yu et al., 2013*; *Raphael et al., 2016*; *Zhan et al., 2018*). For colorectal cancer, the most frequently investigated lag time interval was T29 (time between neoadjuvant chemo-radiotherapy [NACRT] and surgery) (*Foster et al., 2013*; *Wang et al., 2016*; *Petrelli et al., 2016*; *Du et al., 2018*). Only five records considered more than one lag time type (*Seoane et al., 2016*; *Seoane et al., 2012*; *Graboyes et al., 2019*; *van den Bergh et al., 2013*), with the broadest scope considered by Neal et al., who explored 25 unique lag time intervals across 25 cancer types (*Neal et al., 2015*). *Table 1* highlights the distribution of and frequency at which unique lag time intervals were investigated over different cancer sites, and thus synopsizes the distribution of lag time intervals across systematic review and meta-analytical research.

## Findings from systematic reviews

The search strategy and characteristics of each included systematic review are detailed in *Supplementary file 2*. We summarize in *Table 2* their overall characteristics and findings by cancer site and type as well as the corresponding lag time interval.

### Brain

One systematic review investigating the impact of T33 (lag time between surgery and ACT) on overall survival (OS) among patients with brain cancer reported lack of consensus across included studies (*Warren et al., 2019*). Four of 10 included studies reported no association between T33 and OS among patients who experienced T33 >45 days compared to those who experienced T33 <45 days. A further four of 10 included studies reported improved OS among patients who experienced T33 between 31 and 42 days compared to those who experienced T33 <31 days.

### Colorectal

One review reported that higher pathologic complete response (pCR) rates (a prognostic measurement of treatment efficacy in the neoadjuvant setting) were associated with increased tumour downstaging among patients with rectal cancer experiencing time between NACRT and surgery (T29) >6–8 weeks compared to those experiencing T29 <6–8 weeks (*Foster et al., 2013*). In the same review, few included studies demonstrated that T29 >6–8 weeks conferred higher pCR rates. With respect to the impact of prolonged time from diagnosis to surgery (T18) among patients with colon cancer, there was no association between extended T18, regardless of the length, and OS or disease-specific survival (DSS) (*Hangaard Hansen et al., 2018*).

### Eye

Increased time between symptom onset and diagnosis (T4) was associated with increased rates of extraocular disease, metastatic disease, and mortality, but not enucleation, among patients with retinoblastomas (*Mattosinho et al., 2019*).

### Head and neck

A systematic review on oropharyngeal cancers found extended time from diagnosis to surgery (T17) to be associated with poorer OS, shorter time from surgery to initiation of adjuvant radiotherapy (ART) (T30) to be associated with improved overall and recurrence-free survival (RFS), and prolonged time from surgery to completion of ART (T31) to be associated with poorer OS (*Graboyes et al., 2019*).

### Paediatric

Worsened OS was reported among patients with retinoblastomas who experienced longer time between symptom onset and diagnosis (T4) (*Brasme et al., 2012*). Among patients with Ewing's sarcomas, poorer OS due to longer T4 was observed in one review (*Lethaby et al., 2013*), whereas no such association was reported in another review (*Brasme et al., 2012*). While a non-linear association was reported between longer T4 and OS among patients with CNS tumours, where shorter T4 was associated with poorer OS and further extension of T4 was associated with improved OS (*Petrelli et al., 2019*), this association was not observed in the other review (*Brasme et al., 2012*). Both reviews observed no association between extended T4 and OS among patients with osteosarcoma (*Brasme et al., 2012*; *Lethaby et al., 2013*).

**Table 2.** Characteristics of the systematic reviews on the association between time to cancer diagnosis and/or treatment and clinical outcomes, by cancer site/type and lag time interval.

| Cancer | | Lag time | | | | |
|---|---|---|---|---|---|---|
| Site | Type | Interval | Time range | Outcome measures | Overall findings | First author (year) |
| **Brain** | -- | T33 | 15 to >45 days | Overall survival | 4/10 studies: no association between longer time (>45 days) to treatment initiation and overall survival | ***Warren et al., 2019*** |
| | | | | | 4/10 studies: best overall survival was among patients who experienced a moderate time (~31–42 days) to treatment initiation | |
| | | | | | 1/10 studies: a longer time (>45 days) to treatment initiation was associated with poorer overall survival | |
| | | | | | 1/10 studies: improved survival with early treatment initiation (14–21 days) among patients who underwent total resection, and poorer survival for patients who underwent biopsy only | |

*Table 2 continued on next page*

*Table 2 continued*

| Cancer | | Lag time | | | | | First author (year) |
|--------|------|----------|----------|-------------|------------------|-----------------|---------------------|
| Site | Type | Interval | Time range | Outcome measures | Overall findings | | |
| **Colorectal** | | | | | 4/15 studies: higher rates of pathological complete response with longer time intervals (6–8 weeks) between chemoradiotherapy and surgery | | |
| | | | | | 3/15 studies: increased tumour downstaging with longer time intervals (6–8 weeks) | | |
| | Rectal | T29 | <5 days to >12 weeks | Tumour response rate R0 resection Sphincter preservation Surgical complications Disease recurrence | No association between longer time intervals and surgical complication rates, sphincter preservation rates, long-term recurrence rates and survival | | *Foster et al., 2013* |
| | Colon | T18 | 1 to ≥56 days | Overall survival Disease-specific survival Cause-specific survival | 4/5 studies: no association between treatment delay and reduced overall survival regardless of the time intervals investigated | | *Hangaard Hansen et al., 2018* |
| | | | | | 1/5 studies: a clinically insignificant association between longer treatment delay and reduced overall survival | | |
| | | | | | No association between treatment delay and reduced disease-specific survival | | |

*Table 2 continued on next page*

*Table 2 continued*

| Cancer Site | Type | Lag time Interval | Time range | Outcome measures | Overall findings | First author (year) |
|---|---|---|---|---|---|---|
| **Eye** | Retinoblastoma | T4 | 3 to 5 months | Metastasis<br>Mortality<br>Enucleation<br>Extraocular disease | 2/9 studies: association between time to diagnosis (>6 months) and metastatic disease<br><br>2/9 studies: extended time to diagnosis associated with increasing extraocular disease and mortality rates<br><br>No association between time to diagnosis and enucleation | *Mattosinho et al., 2019* |
| **Head and neck** | Oropharyngeal | T17 | 20 to 120 days | Overall survival<br>Disease-specific survival<br>Recurrence-free survival<br>Locoregional control | 9/13 studies: association between longer diagnosis to treatment initiation and poorer overall survival<br><br>4/5 studies: association between shorter time from surgery to postoperative radiotherapy and improved overall survival or recurrence-free survival<br><br>4/5 studies: longer time from surgery to postoperative radiotherapy correlated with poorer overall survival | *Graboyes et al., 2019* |
| | | T30 | >6 to ≥64 days | | | |
| | | T31 | 77 to 100 days | | | |

*Table 2 continued on next page*

*Table 2 continued*

| Cancer | | Lag time | | | | |
|---|---|---|---|---|---|---|
| Site | Type | Interval | Time range | Outcome measures | Overall findings | First author (year) |
| **Paediatric** | | T4 | | | Delayed diagnosis associated with poorer outcomes among patients with retinoblastoma | |
| | | | | | Limited evidence that a delay in diagnosis might be adversely associated with poor oncologic outcomes for patients with leukemia, nephroblastoma, or rhabdomyosarcoma | |
| | Leukemias, lymphomas, brain tumours, neuroblastomas, kidney tumours, soft tissue sarcomas, germ-cell tumours, retinoblastomas | | 2 to 260 weeks | Overall survival Prognostic factors | No association between longer time to diagnosis and oncologic outcomes among patients with osteosarcoma, Ewing's sarcoma, or a central nervous system tumour | *Brasme et al., 2012* |
| | Medulloblastomas, CNS tumours, retinoblastomas, Ewing's sarcomas, bone tumours, osteosarcomas, adenocarcinomas | | 20 to 116 days | Overall survival | Delay in diagnosis associated with poorer survival among patients diagnosed with Ewing's Family of soft tissue sarcomas | *Lethaby et al., 2013* |
| | | | | | Non-linear association between time to diagnosis and survival among patients with central nervous system tumours and non-rhabdomyosarcomas; shortest time to diagnosis associated with poorer survival, however, subsequent extension of time to diagnosis associated with improved survival | |
| | | | | | Time to diagnosis not associated with survival in patients diagnosed with bone tumours | |

*Table 2 continued on next page*

*Table 2 continued*

| Cancer | | Lag time | | | | |
|---|---|---|---|---|---|---|
| Site | Type | Interval | Time range | Outcome measures | Overall findings | First author (year) |
| **Prostate** | -- | T18, T20 | 56 days to 3.7 months | Pathologic characteristics Biochemical recurrence Distant metastasis Overall survival Cause-specific survival | 7/17 studies: no association between time to treatment and poorer oncologic outcomes | ***van den Bergh et al., 2013*** |
| | | | | | 4/17 studies: treatment delay resulted in worse biochemical recurrence rates but no association with overall survival, distant metastasis, or cause-specific survival | |
| | | | | | Prolonged time to treatment (several months or years) does not adversely impact oncologic outcomes in patients with low-risk prostate cancers | |
| | | | | | Limited evidence suggests that prolonged time to treatment might have a negative effect on patients with moderate- and high-risk prostate cancers | |

*Table 2 continued*

| Cancer | | Lag time | | | | |
|---|---|---|---|---|---|---|
| Site | Type | Interval | Time range | Outcome measures | Overall findings | First author (year) |
| | Breast, lung, gastric, oesophageal, gastro-esophageal, pancreatic, hepatocellular, colorectal, prostate, testicular, renal, bladder, upper tract urothelial, cervical, endometrial, ovarian, head and neck, brain/CNS, leukemia, lymphoma, myeloma, connective tissue, carcinoid, thyroid, multisite | T1–T9, T11–T16, T18–T27 | No range of lag times specified | Overall survival Recurrence-free survival Mortality Staging | 142/117 studies: no association between longer delays and poorer outcomes 91/117 studies: positive association between longer delays and poorer outcomes 23/117 studies: negative association between longer delays and poorer outcomes (waiting-time paradox) Some studies found that a longer time to diagnosis and/or treatment was associated with better OS and RFS, while other studies found the opposite. More studies found that shorter times to diagnosis led to better oncologic outcomes in breast, colorectal, head and neck, testicular, and melanoma | *Neal et al., 2015* |
| Multisite | Breast, cervical, colorectal, lung | T10 | 29 to 1092 days | Overall survival Tumour size Tumour stage | Longer wait times associated with a greater risk of poorer clinical outcomes across the breast, cervical, colorectal, and lung cancers Limited evidence confirming specific timeframes during which diagnostic testing should be completed after positive screening test | *Doubeni et al., 2018* |

-- indicates that cancer type was not specified or applicable to the site.

### Prostate

For patients with low-risk prostate cancers, extended time to treatment – either surgery (T18) or radiotherapy (T20) – was not associated with worsened oncologic outcomes or OS (*van den Bergh et al., 2013*), however, some evidence suggests that these associations may be present among patients with moderate- and high-risk prostate cancers.

### Multisite

Two reviews considered the impact of lag times in cancer care across multiple cancer sites (*Neal et al., 2015*; *Doubeni et al., 2018*). One collected evidence on 25 different lag time intervals and 25 different cancer sites *Seoane et al., 2012* and reported consensus across studies on likely associations between shorter times to diagnosis and improved oncologic outcomes among patients with breast, colorectal, head and neck, melanoma, and testicular cancers (*Neal et al., 2015*). The other review concluded that extended time from a positive screening test to diagnosis (T10) among patients with breast, cervical, colorectal, and lung cancers was associated with poorer oncological outcomes such as worsened OS and progressive tumour staging, however, this review emphasized lack of consensus regarding a particular timeframe during which diagnosis should be confirmed after a positive screening test as to mitigate these harmful outcomes (*Doubeni et al., 2018*).

## Findings from meta-analyses

Detailed characteristics (i.e., search strategy, databases searched, number of hits, number of included studies, number of participants, etc.) of each included meta-analysis are presented in *Supplementary file 3*. We summarize, by cancer site and type as well as lag time interval, their methodological characteristics and morbidity- (*Table 3*) and mortality- (*Table 4*) related findings.

## Morbidity-related findings

### Blood

A significant association between shorter lag time between diagnosis and immunotherapy (T21) and decreased disease progression (HR: 0.53, 95% CI [0.33–0.87]) was found among patients with smoldering multiple myeloma (*Zhao et al., 2019*).

### Breast

A meta-analysis investigating the association between time between surgery and ART (T30) among patients with breast cancer reported a significant increase in risk of worsened locoregional control per 1-month increase of T30 (RR: 1.08, 95% CI [1.02–1.14]) (*Gupta et al., 2016*).

### Colorectal

pCR rate was the most common response variable investigated. Higher pCR rates were associated with time from neoadjuvant radiotherapy (NART) and surgery (T28) >4 weeks (RR: 15.71, 95% CI [2.10–117.30]) (*Wu et al., 2018*), as well as with time from NACRT to surgery (T29) >6–8 weeks (RR: 1.42, 95% CI [1.19–1.68]) (*Petrelli et al., 2016*), >7–8 weeks (RR: 1.45, 95% CI [1.18–1.78]) (*Wang et al., 2016*), and ≥8 weeks (RR: 1.24, 95% CI [1.14–1.35]) (*Du et al., 2018*).

### Head and neck

One meta-analysis investigated the association between time from symptom onset to (1) first being seen by a primary care provider (PCP) (T1), (2) referral to a specialist (T2), and (3) diagnosis (T4), as well as time from first being seen by a PCP to diagnosis (T11) on TNM staging among patients with oral cancer (*Seoane et al., 2016*). There was greater risk of worsened TNM staging (based on 'short' and 'long' lag time cut-offs determined by included studies) associated with longer T1 (RR: 1.55, 95% CI [1.14–2.12]), T11 (RR: 2.15, 95% CI [1.08–4.29]), and any lag time (T1, T2, T4, or T11) (RR: 1.66 [1.25–2.20]). Another meta-analysis assessing the impact of time from symptom onset to diagnosis (T4) on TNM staging among patients with advanced-stage oral and oropharyngeal cancers found greater odds of increased TNM staging among patients who experienced T4 classified as 'long' by included original studies (OR: 1.32, 95% CI [1.07–1.62]) (*Gómez et al., 2009*).

**Table 3.** Morbidity-related findings of meta-analyses on the association between time to cancer diagnosis and/or treatment and clinical outcomes, by cancer site/type and lag time interval.

| Cancer | | Lag time interval | | | | Findings | | First author (year) |
|---|---|---|---|---|---|---|---|---|
| Site | Type | Interval | Type | Comparison | Time range | Outcome measures | Pooled risk estimate [95% CI] (model type, heterogeneity statistics $I^2$ or Ri) | |
| **Blood** | Smoldering multiple myeloma | T21 | Categorical | No distinct cut-off specified* | No range of lag times specified | Disease progression | HR: 0.53 [0.33–0.87] (random-effects, $I^2$ = 86%) | *Zhao et al., 2019* |
| | | | | | | Therapy response rate | HR: 0.87 [0.73–1.03] (fixed-effects) | |
| **Breast** | -- | T30 | Continuous | Per 1-month increase | 31 to 203 days | LR | RR: 1.08 [1.02–1.14] (fixed-effects) | *Gupta et al., 2016* |

*Table 3 continued on next page*

Table 3 continued

| Cancer | | Lag time interval | | | | Findings | | First author |
|---|---|---|---|---|---|---|---|---|
| Site | Type | Interval | Type | Comparison | Time range | Outcome measures | Pooled risk estimate [95% CI] (model type, heterogeneity statistics I² or Ri) | (year) |
| **Colorectal** | | | | | | | | |
| | | | | | | pCR rate | RR: 15.71 [2.10–117.30] (fixed-effects) | |
| | | | | | | Downstaging rate | RR: 2.63 [1.77–3.90] (fixed-effects) | |
| | | | | | | TNM stage | RR: 1.49 [1.23–1.81] (fixed-effects) | |
| | | | | | | Sphincter-preserving rate | RR: 1.05 [0.96–1.15] (fixed-effects) | |
| | | | | | | R0 resection rate | RR: 1.08 [0.99–1.19] (fixed-effects) | |
| Rectal | | T28 | Categorical | >4 vs. <4 weeks | 5 days to 8 weeks | Incidence of postoperative complications | RR: 0.81 [0.70–0.95] (fixed-effects) | *Wu et al., 2018* |
| Rectal | | T29 | Categorical | >7–8 vs. <7–8 weeks | 5 to >12 weeks | pCR rate | RR: 1.45 [1.18–1.78] (fixed-effects) | *Wang et al., 2016* |
| | | | Categorical | >6–8 vs. <6–8 weeks | 4 to 14 weeks | pCR rate | RR: 1.42 [1.19–1.68] (fixed-effects) | *Petrelli et al., 2016* |
| | | | Categorical | ≥8 vs. <8 weeks | 4 to 14 weeks | pCR rate | RR: 1.24 [1.14–1.35] (random-effects, I² = 9.8%) | *Du et al., 2018* |
| | | | | | | Operative time | SMD: 0.15 [0.03–0.32] (random-effects, I² = 24.3%) | |
| | | | | | | Incidence of LR | RR: 0.92 [0.61–1.37] (random-effects, I² = 65.1%) | |
| | | | | | | Postoperative complications | RR: 0.95 [0.83–1.09] (random-effects, I² = 25.6%) | |
| | | | | | | Anastomotic leakage | RR: 0.89 [0.49–1.63] (random-effects, I² = 0%) | |
| | | | | | | Sphincter-preserving surgery | RR: 0.99 [0.91–1.07] (random-effects, I² = 0%) | |

*Table 3 continued on next page*

*Table 3 continued*

| Cancer | | Lag time interval | | Findings | | | | First author (year) |
|---|---|---|---|---|---|---|---|---|
| Site | Type | Interval | Type | Comparison | Time range | Outcome measures | Pooled risk estimate [95% CI] (model type, heterogeneity statistics $I^2$ or Ri) | |
| | | | T1 | Categorical | No distinct cut-off specified | >30 days to >1 month | TNM staging | RR: 1.55 [1.14–2.12] (fixed-effects) |
| | | | | | | | | RR: 1.55 [1.14–2.12] (random-effects, Ri = 0.00) |
| | | | T11 | Categorical | No distinct cut-off specified | >30 days to >1 month | TNM staging | RR: 1.83, [1.31–2.56] (fixed-effects) |
| | | | | | | | | RR: 2.15 [1.08–4.29] (random-effects, Ri = 0.74) |
| | Oral | T1, T2, T4, T11 | Categorical | No distinct cut-off specified | >30 to >45 days | TNM staging | RR: 1.61 [1.33–1.93] (fixed-effects) | Seoane et al., 2016 |
| | | | | | | | | RR: 1.66 [1.25–2.20] (random-effects, Ri = 0.49) |
| | Oropharyngeal, Oral (advanced stage) | | T4 | Categorical | No distinct cut-off specified | No range of lag times specified | TNM staging | OR: 1.32 [1.07–1.62] (fixed-effects) |
| | | | | | | | | OR: 1.25 [0.84–1.85] (random-effects, Ri = 0.70) |
| | | | | | | | pCR rate | OR: 0.97 [0.73–1.30] (fixed-effects) |
| | | | | | | | Postoperative mortality | OR: 0.75 [0.40–1.44] (fixed-effects) |
| | | | | | | | Anastomotic leakage | OR: 1.33 [0.69–1.85] (fixed-effects) | Gómez et al., 2009 |
| Head and neck | Esophageal | T29 | Categorical | >7–8 vs. ≤7–8 weeks | ≤46 to >64 days | R0 resection rate | OR: 1.71 [1.14–2.22] (fixed-effects) | Lin et al., 2016 |

Significant pooled risk estimates are bolded.

-- indicates that cancer type not specified or applicable to the site.

*Meta-analysis utilized the 'early' and 'late' lag time interval definitions in included studies without standardization of lag time cut-offs.

CI = confidence interval. HR = hazard ratio. LR = local recurrence. OR = odds ratio. pCR = pathological complete response. RR = risk ratio. SMD = standard mean difference.

**Table 4.** Mortality-related findings of meta-analyses on the association between time to cancer diagnosis and/or treatment and clinical outcomes, by cancer site/type and lag time interval.

| Cancer | | Lag time interval | | | | Findings | | |
| --- | --- | --- | --- | --- | --- | --- | --- | --- |
| Site | Type | Interval | Type | Comparison | Time range | Outcome measures* | Pooled risk estimate [95% CI] (model type, heterogeneity statistics $I^2$ or Ri) | First author (year) |
| Blood | Smoldering multiple myeloma | T21 | Categorical | No distinct cut-off specified† | No range of lag times specified | Mortality | HR: 0.90 [0.72–1.12] (fixed-effects) | Zhao et al., 2019 |
| Brain | Glioblastoma | T30 | Continuous | Per 1-week increase | 12 to 53 days | Mortality | HR: 0.98 [0.90–1.08] (non-adjusted model) | Loureiro et al., 2016 |
| Breast | -- | T30 | Continuous | Per 1-month increase | 31 to 203 days | Mortality | RR: 0.99 [0.94–1.05] (fixed-effects) | Gupta et al., 2016 |
| | | T32 | Continuous | Per 4-week increase | <21 days to >3 months | Mortality | HR: 1.15 [1.03–1.28] (random-effects, $I^2$ = 75.4%) | |
| | | | | | | Worsened DFS | HR: 1.16 [1.01–1.33] (fixed-effects) | Yu et al., 2013 |
| | -- | | Continuous | Per 4-week increase | <21 days to >3 months | Mortality | RR: 1.04 [1.01–1.08] (fixed-effects) | |
| | | | | | | | RR: 1.08 [1.01–1.15] (random-effects, $I^2$ = 60%) | |
| | | | | | | | RR: 1.05 [1.01–1.08] (fixed-effects) | |
| | | | | | | Worsened DFS | RR: 1.05 [1.01–1.10] (random-effects, $I^2$ = 94.9%) | Raphael et al., 2016 |
| | -- | | Continuous | Per 4-week increase | <21 days to >3 months | Mortality | HR: 1.13 [1.08–1.19] (random-effects, $I^2$ = 78.9%) | Liu et al., 2017 |
| | | | | | | Worsened DFS | HR: 1.14 [1.05–1.24] (random-effects, $I^2$ = 60.9%) | |

*Table 4 continued on next page*

*Table 4 continued*

| Cancer | Lag time interval | | | | Findings | | |
|---|---|---|---|---|---|---|---|
| Site | Interval | Type | Comparison | Time range | Outcome measures* | Pooled risk estimate [95% CI] (model type, heterogeneity statistics I² or RI) | First author (year) |
| **Colorectal** | | | | | | | |
| Rectal | T28 | Categorical | >4 vs. <4 weeks | 5 days to 8 weeks | Mortality | RR: 0.75 [0.53–1.07] (random-effects, I² = 60%) | *Wu et al., 2018* |
| | | | | | Worsened DFS | RR: 0.78 [0.84–1.14] (fixed-effects) | |
| Rectal | | Categorical | >6–8 vs. <6–8 weeks | 4 to 14 weeks | Mortality | RR: 0.85 [0.50–1.43] (random-effects, I² = 59%) | *Petrelli et al., 2016* |
| | | | | | Worsened DFS | RR: 0.81 [0.58–1.12] (random-effects, I² = 61%) | |
| Rectal | T29 | Categorical | ≥8 vs. <8 weeks | 4 to 14 weeks | Mortality | RR: 0.98 [0.91–1.06] (random-effects, I² = 42.4%) | *Du et al., 2018* |
| | | | | | Worsened DFS | RR: 1.04 [0.94–1.14] (random-effects, I² = 46.7%) | |
| | T32 | | | | Mortality | RR: 1.20 [1.15–1.26] (fixed-effects) | |
| Colorectal (Stage II/III) | | Categorical | >8 vs. <8 weeks | 4 to 8+ weeks | Worsened RFS | RR: 0.98 [0.89–1.08] (fixed-effects) | *Des Guetz et al., 2010* |
| Colorectal (Stage II/III) | | Continuous | Per 4-week increase | 4 to >36 weeks | Mortality | HR: 1.14 [1.10–1.17] (fixed-effects) | *Biagi et al., 2011* |
| | | | | | DFS | HR: 1.14 [1.10–1.18] (fixed-effects) | |
| | | Categorical | >6–8 vs. <6–8 weeks | <4 to >12 weeks | Mortality | HR: 1.20 [1.04–1.38] (fixed-effects) | *Petrelli et al., 2019* |
| Gastric | | | | | Mortality | HR: 1.41 [0.94–1.28] (random-effects, I² = 90%) | |
| | | | | | Mortality | HR: 1.27 [1.21–1.33] (fixed-effects) | |
| Colorectal | | | | | Mortality | HR: 1.27 [1.25–1.28] (random-effects, I² = 70%) | |
| Pancreatic | | | | | Mortality | HR: 1.00 [1.00–1.01] (fixed-effects) | |

*Table 4 continued on next page*

*Table 4 continued*

| Cancer | | Lag time interval | | | | Findings | | First author (year) |
|---|---|---|---|---|---|---|---|---|
| Site | Type | Interval | Type | Comparison | Time range | Outcome measures* | Pooled risk estimate [95% CI] (model type, heterogeneity statistics I² or Ri) | |
| **Head and neck** | | | | | | | | |
| | | T1 | Categorical | No distinct cut-off specified | 30 to 60 days | Mortality | **RR: 1.54 [1.21–1.94]** (fixed-effects) | |
| | | | | | | | RR: 1.67 [0.88–3.19] (random-effects, Ri = 0.85) | |
| | | T2 | Categorical | No distinct cut-off specified | 72 days | Mortality | **RR: 2.72 [1.45–5.09]** (fixed-effects) | |
| | | | | | | | **RR: 3.17 [1.12–9.00]** (random-effects, Ri = 0.61) | |
| | | T4 | Categorical | No distinct cut-off specified | 108 to 180 days | Mortality | **RR: 1.04 [1.01–1.07]** (fixed-effects) | |
| | | | | | | | **RR: 1.04 [1.01–1.07]** (random-effects, Ri = 0.00) | |
| | | T11 | Categorical | No distinct cut-off specified | 21 to 106 days | Mortality | **RR: 1.34 [1.00–1.78]** (fixed-effects) | |
| | | | | | | | RR: 1.32 [0.66–2.66] (random-effects, Ri = 0.82) | |
| | -- | T1, T2, T4, T11 | Categorical | No distinct cut-off specified | 21 to 180 days | Mortality | **RR: 1.05 [1.02–1.07]** (fixed-effects) | |
| | | | | | | | **RR: 1.34 [1.12–1.61]** (random-effects, Ri = 0.95) | *Seoane et al., 2012* |
| | | T2 | Categorical | No distinct cut-off specified | >1 month | Mortality | **RR: 2.48 [1.39–4.42]** (fixed-effects) | |
| | | | | | | | **RR: 2.48 [1.39–4.42]** (random-effects, Ri = 0.00) | |
| Oral | | T1, T2, T4, T11 | Categorical | No distinct cut-off specified | >30 to >45 days | Mortality | RR: 1.02 [0.93–1.12] (fixed-effects) | |
| | | | | | | | **RR: 1.35 [0.84–2.18]** (random-effects, Ri = 0.94) | *Seoane et al., 2016* |
| Esophageal | | T29 | Categorical | >7–8 vs. ≤7–8 weeks | ≤46 to >64 days | Mortality, 2 years | **OR: 1.40 [1.09–1.80]** (fixed-effects) | *Lin et al., 2016* |
| | | | | | | Mortality, 5 years | OR: 1.14 [0.84–1.54] (fixed-effects) | |

*Table 4 continued on next page*

*Table 4 continued*

| Cancer | | Lag time interval | | | | Findings | | |
|---|---|---|---|---|---|---|---|---|
| Site | Type | Interval | Type | Comparison | Time range | Outcome measures* | Pooled risk estimate [95% CI] (model type, heterogeneity statistics $I^2$ or Ri) | First author (year) |
| | | | Categorical | No distinct cut-off specified | | Mortality | **HR: 1.18 [1.06–1.32] (random-effects, $I^2$ = 17.6%)** | ***Liu et al., 2017*** |
| | -- | | Continuous | Per 1-week increase | <15 days to >12 weeks | Mortality | HR: 1.04 [1.00–1.09] (random-effects, $I^2$ = 9.05%) | |
| **Ovarian** | -- | T32 | Categorical | No distinct cut-off specified | 19 to 42 days | Mortality, 3 years | OR: 1.06 [0.90–1.24] (random-effects, $I^2$ = 64.3%) | ***Usón et al., 2017*** |

Significant pooled risk estimates are bolded.

-- indicates that cancer type not specified or applicable to the site.

*Response variables of interest indicate the directionality of the pooled risk estimate (e.g., RR >1 associated with greater risk of mortality among patients with lag time intervals to cancer care endpoint greater than the lag time cut-off considered by the meta-analysis).

†Meta-analyses utilized the 'early' and 'deferred' lag time interval definitions in included studies without standardization of lag time cut-offs.

CI = confidence interval. DFS = disease-free survival. HR = hazard ratio. $I^2$ = heterogeneity. OR = odds ratio. RFS = recurrence-free survival. Ri = proportion of total variance due to between-study variance. RR = risk ratio. SMD = standardized mean difference.

### Mortality-related findings

#### Blood
No association was found between extended time from diagnosis to immunotherapy (T21) and risk of mortality among patients with smoldering multiple myelomas (*Zhao et al., 2019*).

#### Brain
Time (per 1-week increase) between surgery and NART (T30) was not associated with an increased risk of mortality among patients with brain cancers (*Loureiro et al., 2016*).

#### Breast
No change in risk of mortality was found per 1-month increase in the time between surgery and ART (T30) (*Gupta et al., 2016*). Three meta-analyses reported a significant increase in risk of mortality per 1-month increase in T32 (time between surgery and ACT) (HR: 1.15, 95% CI [1.03–1.28], *Yu et al., 2013*; RR: 1.04, 95% CI [1.01–1.08], *Raphael et al., 2016*; and HR: 1.13, 95% CI [1.08–1.19], *Zhan et al., 2018*) as well as risk of worsened DFS per 1-month increase in T32 (HR: 1.16, 95% CI [1.01–1.33], *Yu et al., 2013*; RR: 1.05, 95% CI [1.01–1.10], *Raphael et al., 2016*; HR: 1.14, 95% CI [1.05–1.24], *Zhan et al., 2018*).

#### Colorectal
One meta-analysis investigated the impact of time between NART and surgery (T28) >4 weeks on mortality and risk of worsened DFS and found no association (*Wu et al., 2018*). Two meta-analyses found no association between undergoing surgery >6–8 weeks (*Petrelli et al., 2016*) or ≥8 weeks (*Du et al., 2018*) after NACRT (T29) and risk of mortality as well as risk of worsened DFS. Three other meta-analyses evaluated the impact of time between surgery and ACT (T32) on mortality among patients with colorectal cancer (*Petrelli et al., 2019*; *Des Guetz et al., 2010*; *Biagi et al., 2011*). Those that classified the length of T32 categorically reported an increased risk of mortality among patients with colorectal cancer who experienced T32 >6–8 weeks (HR: 1.27, 95% CI [1.25–1.28]) (*Petrelli et al., 2019*) and >8 weeks (RR: 1.20, 95% CI [1.15–1.26]) (*Des Guetz et al., 2010*). Similarly, a greater risk of mortality per 4-week increase in T32 was reported (HR: 1.14, 95% CI [1.10–1.17]) (*Biagi et al., 2011*).

#### Head and neck
Two meta-analyses investigated the influence of time from symptom onset to (1) first being seen by a PCP (T1), (2) referral for diagnosis (T2), and (3) diagnosis (T4), and from first being seen by a PCP to diagnosis (T11), with one considering all head and neck cancers and the other oral cancers only (*Seoane et al., 2016*; *Seoane et al., 2012*). For all head and neck cancers, significantly greater risks of mortality were found among patients who experienced extended T1 (RR: 1.54, 95% CI [1.21–1.94]), T2 (RR: 3.17, 95% CI [1.12–9.00]), T4 (RR: 1.04, 95% CI [1.01–1.07]), and T11 (RR: 1.34, 95% CI [1.00–1.78]) (*Seoane et al., 2012*). Restriction to oral cancers revealed a greater risk of mortality among patients who experienced extended T2 (*Seoane et al., 2016*). A meta-analysis investigating the effect of time from NACRT to surgery on mortality (T29) among patients with esophageal cancer found significantly greater odds of 2-year mortality among patients who experienced T29 >7–8 weeks, with no association found between T29 >7–8 weeks and the odds of 5-year mortality (*Lin et al., 2016*).

#### Ovarian
Of the two meta-analyses that evaluated the association between time from surgery to ACT (T32) and mortality among patients with ovarian cancer, one found a significantly greater risk of mortality per 1-week increase in T32 (HR: 1.04, 95% CI [1.00–1.09]) or among those who experienced extended T32 (HR: 1.18, 95% CI [1.06–1.32]) (*Liu et al., 2017*). The other meta-analysis reported no association between extended T32 and odds of 3-year mortality (*Usón et al., 2017*).

## Discussion
We conducted a scoping review of systematic reviews and/or meta-analyses in order to characterize the body of pre-pandemic evidence on the known associations between lag time in cancer care and

control and patient outcomes. Our comprehensive overview of the available peer-reviewed literature enabled the identification of either consistency or disagreement across reviews on the same lag time, cancer site, and outcome. Select comparisons across included meta-analyses investigating the same associations uncovered varying approaches to quantifying the effect of lag time on oncologic outcomes. Specifically, four lag time intervals for different cancer sites provided informative perspectives: (1) from surgery to ACT (T32) for breast cancer, (2) from NACRT to surgery (T29) for rectal cancer, (3) from surgery to ACT (T32) for colorectal cancer, and (4) from surgery to ACT (T32) for ovarian cancer. These comparisons revealed overarching methodological gaps in lag time literature.

## T32 and breast cancer

The three meta-analyses investigating the effect of extended time to ACT (T32) on mortality among patients with breast cancer reported a significantly greater risk of mortality per 1-month extension of T32 (*Yu et al., 2013*; *Raphael et al., 2016*; *Zhan et al., 2018*). Although they provided consistent conclusions, differences in the magnitude of the reported risk estimates could be attributed to conflicting inclusion of a registry-based study (*Hershman et al., 2006b*) that found a significant association between extended T32 and mortality. This registry-based study (*Hershman et al., 2006b*) was included in *Yu et al., 2013* and *Zhan et al., 2018* meta-analyses, which yielded risk estimates greatest in magnitude, and accounted for 21.24% and 8.09% of the weight in their analyses, respectively. *Raphael et al., 2016*, who reported the most conservative pooled risk estimate, did not include this registry-based study (*Hershman et al., 2006b*). *Raphael et al., 2016* argued that inclusion of this registry-based study in Yu et al.'s analysis (*Yu et al., 2013*) could have introduced bias by confounding or misclassification of ACT as palliative rather than curative, resulting in an overestimation of risk associated with prolonged T32. By extension, this argument could also apply to Zhan et al.'s analysis (*Zhan et al., 2018*). However, Raphael et al.'s argument relied on the assumption that those who received palliative ACT were not only less likely to survive, but also more likely to experience longer time to ACT after surgery, thus inducing an artifactual association between shorter time from surgery to ACT and mortality. Because it is unknown whether patients undergoing palliative care would be considered lower priority for receiving ACT or, conversely, experience shorter time to ACT due to directed resources specific to palliative care units, it is unclear as to whether the possible inclusion of patients undergoing palliative ACT in the registry-based study (*Hershman et al., 2006b*) would have led to an over- or under-estimation of *Yu et al., 2013* and *Zhan et al., 2018* risk estimates.

## T29 and rectal cancer

The three meta-analyses evaluating the impact of time from NACRT to surgery (T29) on pCR rates among patients with advanced rectal cancer provided different conclusions regarding the optimal timing of surgery (*Wang et al., 2016*; *Petrelli et al., 2016*; *Du et al., 2018*). Each reported protective associations of extended T29 for pCR rates using different cut-offs to define 'longer' versus 'control' T29: 7 weeks (*Wang et al., 2016*), 6–8 weeks (*Petrelli et al., 2016*), and 8 weeks after NACRT (*Du et al., 2018*). These cut-offs were determined based on the seminal Lyon R90-01 trial, which investigated the impact of 6–8 weeks between NART and sphincter-preserving surgery compared to 2 weeks on pathological downstaging among patients with rectal cancer and established 6–8 weeks as an accepted lag time of NART or NACRT after surgery in clinical practice (*Francois et al., 1999*). However, an evaluation of a 6- to 8-week length of time between preoperative radiotherapy and surgery in the trial might not have precluded the potential for shorter time between preoperative chemoradiotherapy and surgery (T29) (>2 and <6–8 weeks), or longer T29 (>6–8 weeks) to sustain benefit for patients with rectal cancer.

Considering this uncertainty, Wang et al. conducted subgroup analyses based on different cut-offs of T29 (5, 6, 7, 8, 10, and 12 weeks) (*Supplementary file 3*) and found that performing surgery at 7 and 8 weeks yielded a significantly improved pCR rates compared to performing surgery earlier, at 5 or 6 weeks, or later, at 10 or 12 weeks (*Wang et al., 2016*). This reported 'optimal window', which is narrower than the clinically accepted 6- to 8-week window, suggests that benefits associated with NACRT for patients with advanced rectal cancer are dependent on the timing of surgery and also on inter-individual variation in response to NACRT. Among rectal cancer patients, surgery performed too soon may not allow for the maximal anti-tumourigenic response to NACRT, however, waiting too

long could mitigate any potential benefit that could be maintained from NACRT and permit tumour repopulation.

## T32 and colorectal cancer

Three meta-analyses (*Petrelli et al., 2019*; *Des Guetz et al., 2010*; *Biagi et al., 2011*) investigating the association between time from surgery to ACT on mortality among patients with colorectal cancer reported significant associations between extended time from surgery to ACT (T32) and risk of mortality. Yet, all three *Petrelli et al., 2019*; *Des Guetz et al., 2010*; *Biagi et al., 2011* used differing statistical methods with regard to lag time variable classification. *Des Guetz et al., 2010* and *Petrelli et al., 2019* considered lag time categorically with respective cut-offs of >8 and >6–8 weeks, while *Biagi et al., 2011* considered lag time as a continuous variable. *Biagi et al., 2011* excluded four studies (*Berglund et al., 2008*; *André et al., 2007*; *Gray et al., 2007*; *Taal et al., 2001*) that were included in Des Guetz et al.'s analysis (*Raphael et al., 2016*) based on lack of adjustment for potential confounders. Two studies (*Cheung et al., 2009*; *Hershman et al., 2006a*) included in *Biagi et al., 2011* and *Des Guetz et al., 2010* analysis, which cumulatively contributed 69.05% of the weight in Biagi et al.'s analysis (weight was not reported by Des Guetz et al.), reported risk estimates highly similar to those yielded from the two meta-analyses. After exclusion of the two largest studies (*Cheung et al., 2009*; *Hershman et al., 2006a*) in both analyses, significance was maintained (*Supplementary file 3*).

## T32 and ovarian cancer

The two meta-analyses on the association between time to ACT after surgery (T32) and risk of mortality among patients with ovarian cancer reported distinct findings; one found a significantly increased risk of mortality due to longer T32 (*Liu et al., 2017*), whereas the other did not find this relationship significant (*Usón et al., 2017*). Key methodological differences could account for these conflicting findings. Although there was substantial overlap of included studies between the meta-analyses, the one that found an increased risk of mortality due to longer T32 (*Liu et al., 2017*) stratified the analysis by lag time variable type (categorical or continuous), while the other meta-analysis that found no association (*Usón et al., 2017*) included studies independent of the lag time variable type. Inclusion of studies in the same analysis independent of or dependent upon how lag time was considered likely contributed to the substantial heterogeneity observed in the meta-analysis that did not stratify by lag time variable type ($I^2$ = 64.3%) (*Usón et al., 2017*) compared to the one that did ($I^2$ = 17.6%, 9.05%) (*Liu et al., 2017*). Notably, the cut-offs defining 'early' versus 'deferred' lag time intervals in both meta-analyses were based on those used in the included original studies. It is also likely that the wide range of lag time intervals across studies might have influenced the precision and significance of the reported risk estimates, hence the importance of standardization of lag time cut-offs in meta-analyses and the impact of lag time variable type on statistical findings.

## Methodological gaps in lag time literature

In juxtaposing the design and analytical approaches of included reviews, we identified three predominant methodological gaps in lag time literature which can guide future research on lag times in cancer care. Firstly, consideration of change in intervention modality over time was inconsistent across included reviews. Failure to account for improvement of clinical protocols and treatment regimens, which can confer greater protection against morbidity- and mortality-related outcomes, could have biased the observed risk estimates based on recency of included studies and their corresponding weight in the meta-analyses. For example, the three meta-analyses (*Yu et al., 2013*; *Raphael et al., 2016*; *Zhan et al., 2018*) that reported significant associations between time to ACT after surgery (T32) and mortality among patients with breast cancer included studies for which chemotherapy regimens were anthracycline-based and/or CMF regimens with the earliest included study dating to 1989 (*Pronzato et al., 1989*). This could constrain generalizability of results to patients treated with more recent ACT regimens which primarily include taxanes as standard care (*Zaheed et al., 2019*). It is noteworthy, however, that apart from the recent advent of immunotherapy and targeted therapies, which fall under the umbrella of precision medicine, there have not been major paradigm shifts in treatment provision during the timespan covered by our search (1 January 2010 to 31 December 2019). Nevertheless, our restriction on publication date of included systematic reviews and/or meta-analyses does not prevent the inclusion of studies that evaluate interventions that have since evolved

in type, dosing, and/or timing. Indeed, future ubiquity of novel therapies under the purview of precision medicine will necessitate consideration of the evolution of cancer care in meta-analytical lag time research. In such cases, conducting sensitivity analyses based on treatment type, regimen, or date of publication can provide a clearer understanding of the external validity of reported risk estimates. The findings of select included meta-analyses that did conduct such sensitivity analyses (*Zhao et al., 2019*; *Zhan et al., 2018*; *Liu et al., 2017*; *Supplementary file 3*) demonstrated the variation in significance of the association between lag time and clinical outcomes due to treatment specification and recency of publication in addition to baseline patient characteristics, such as demographics, stage, date of administration, dosing, type of therapy, and clinical decisions.

Secondly, ambiguity in defining lag time interval start and endpoints in the reviews and original studies could lead to exposure misclassification. For example, in the case of categorical lag time variables, specification of whether an endpoint is considered as initiation of NACRT, completion of NACRT, or time of medical charting of NACRT, could influence the length of the lag time interval measured and classification of such exposure as experimental or control.

The Aarhus statement, a set of recommendations and checklist resulting from discussion regarding methodological concerns in research conducted on lag times to cancer diagnosis, identified unclear definitions of start and endpoints in lag times, as well as inconsistency in defining these points across studies evaluating the same lag time as primary obstacles to aggregating research on lag time to cancer diagnosis (*Weller et al., 2012*). We argue that the same concerns would apply to research on lag time between any start and endpoint on the cancer care continuum, not only within time to cancer diagnosis.

Some of the included qualitative systematic reviews initially intended to meta-analyze the abstracted data (*Hangaard Hansen et al., 2018*; *Graboyes et al., 2019*; *Foster et al., 2013*; *Lethaby et al., 2013*), however, variability in start and endpoint definitions of lag times across included studies and in type of lag time variable (categorical vs. continuous) prevented researchers from pursuing quantitative analyses. This lack of clarity of exposure definitions stymies lag time research and calls for an expansion of the pre-existing Aarhus statement (*Weller et al., 2012*) to encompass lag time research across the entire cancer care continuum. Refinements of the Aarhus statement integrating suggestions from clinicians and cancer diagnosis researchers have been proposed (*Coxon et al., 2018*), yet are still restricted to cancer diagnosis, rather than applicable to the entire cancer care continuum. Additions to the Aarhus statement regarding the entire continuum could provide not only recommendations for standardizing lag time definitions in randomized controlled trials and observational studies, but also for justifying classification of lag time categorically versus continuously (e.g., evaluation of a lag time cut-off in standard care). Apart from residual confounding that can arise from categorical classification of lag time, the method of defining lag time categories using cut-offs can substantially impact resulting risk estimates and their corresponding uncertainty. Some included meta-analyses (*Seoane et al., 2016*; *Seoane et al., 2012*; *Zhao et al., 2019*; *Gómez et al., 2009*; *Usón et al., 2017*; *Liu et al., 2017*) utilized categories of 'early' and 'deferred' care defined by original studies. Even when classifying lag time categorically is appropriate, not standardizing cut-offs across included studies can give rise to more significant variation in risk estimates and constrict interpretation of resulting findings. Overall, recommendations for defining lag time exposure across the entire continuum and strategizing statistical classification of lag time exposure is necessary for improving the quality and utility of future lag time research.

Thirdly, confounding by indication needs to be taken into consideration in research conducted on lag times in cancer care. Otherwise monikered the 'waiting-time paradox', confounding by indication is the implication that patients with more advanced-stage disease are prioritized for referral and subsequent treatment within a given health system more rapidly than patients with early-stage, less severe disease. The overall effect of this prioritization leads to the indication that early referral and/or treatment is associated with higher mortality; this survival trend is in direct contradiction with the typical log-linear approach taken in meta-analyses on the association between extended lag time to cancer treatment and risk of mortality. The presence of the waiting-time paradox in meta-analyses can be most clearly identified when lag time is treated as a continuous rather than categorical variable. However, this can be difficult as studies included in meta-analyses often regard lag times as dichotomous exposures. Few of the meta-analyses included in the current scoping review reported risk estimates for lag time intervals as a continuous exposure (*Gupta et al., 2016*; *Loureiro et al., 2016*;

*Yu et al., 2013*; *Raphael et al., 2016*; *Zhan et al., 2018*; *Biagi et al., 2011*; *Liu et al., 2017*). Notwithstanding appropriate justification for treating lag time categorically, either sensitivity analyses using different cut-offs of categorical lag time or meta-analyses using continuous lag time could facilitate identification of the waiting-time paradox and provide insights regarding its mechanisms.

Apart from inherent limitations of the included systematic reviews and meta-analyses, some over-arching limitations in the methodology of our scoping review need to be acknowledged. Restricting our search to systematic reviews and/or meta-analyses published after 2010 did not prevent the introduction of medical interventions that have changed over time in included literature, however, our reporting of stratified analyses performed in included meta-analyses, when available, by treatment specifications (e.g., dosing, chemotherapy type) does provide context regarding modification of lag time on clinical outcomes by intervention modality. Further, even though some meta-analyses evaluated the same parameters, variability in statistical methods limited the scope of our comparisons between these meta-analyses. While we did not conduct quality assessment due to the nature of the included literature, we did describe characteristics of each review which could aid in assessing the validity and generalizability of each review's findings. Moreover, differences in reporting across included systematic reviews and meta-analyses prevented our ability to provide (1) aggregated summary statistics of the associations between lag time and oncologic outcomes, and (2) the distribution of lag times experienced by patients included in each study within each meta-analysis. Our inability to provide these statistics stresses the need for transparent and comprehensive reporting in forthcoming research on lag times in cancer care. Finally, despite the importance of examining treatment variations among patients within clinical trial settings, which are circumscribed to well-established rules and procedures, real-world evidence, such as data from electronic medical or health records, can provide further insight into patient profiles, treatment choice, drug adherence, and adverse event management. These individual-level factors, which are not captured by the included meta-analyses, can appreciably influence oncologic outcomes and thus the true causal effect of lag time on morbidity- and mortality-related outcomes.

## Contextualizing findings of the scoping review amidst the COVID-19 pandemic

Our comprehensive map of lag time intervals and clinical outcomes across multiple cancer sites is intended to serve as a reference point for future research evaluating the pandemic's impact on lag times in cancer control and care. As it can take years for cancer-related survival outcomes to accrue, it is too soon to accurately quantify the impact of extended times to diagnosis and treatment attributable to the pandemic. Hence, the perspectives presented herein on the impact of lag time in cancer control and care not only aid in providing a contextual reference for pandemic-induced lag time compared to standard-of-care lag time experienced prior to the pandemic, but also inform ongoing research on these unprecedented lag times experienced by patients. The reported risk estimates therefore represent the estimated associations between lag time and cancer outcomes prior to the pandemic, which can serve as a standard-of-care reference of these associations regardless of other changes to cancer care due to the COVID-19 pandemic.

Recently published modelling studies simulating the long-term impact of pandemic-induced lag times on cancer-related outcomes have informed the degree of tolerability of the widespread consequences of the pandemic on cancer care systems (*Maringe et al., 2020*; *Sud et al., 2020*). A UK population-based modelling study predicting the impact of lag times to diagnosis of colorectal, breast, lung, and esophageal cancers on survival during the 12 months after national lockdown measures began estimated increases of 7.9–9.6%, 15.3–16.6%, 4.8–5.3%, and 5.8–6.0% in deaths due to breast, colorectal, lung, and esophageal cancers, respectively, up to five years after diagnosis compared to pre-pandemic data (*Maringe et al., 2020*). Another modelling study from our group projected a 2% increase in cancer deaths, or an excess 355,172 life-years-lost, in Canada between 2020 and 2030 due to pandemic-induced lag times to both cancer diagnosis and treatment, assuming the cancer care system returned to normal capacity in 2021 (*Malagón et al., 2022*). Triage systems prioritizing patients into treatment and diagnostic systems (*Farah et al., 2021*), on top of system-related constraints, have resulted in unprecedented lag times that exceed those experienced pre-pandemic (*Jazieh et al., 2020*). This evokes the question as to whether adverse consequences associated with such pandemic-related lag times will be harsher than those described in retrospective data.

The prevailing concern is that any attenuation of the impact of pandemic-induced lag time on cancer diagnosis and treatment is dependent on immediate recovery of cancer care infrastructure. As was seen most recently with the Omicron variant, which stressed already over-burdened healthcare systems worldwide, healthcare systems' recovery to full capacity is tenuous. Such fragility signals for an ongoing need to quantify and contextualize lag time's impact on cancer-related outcomes. This need was highlighted by Hanna et al.'s meta-analysis, which reported generalizable measures of effect of 4-week lag time to treatment, stratified by modality, across seven common cancers (*Hanna et al., 2020*). While meta-analyses, especially the one by *Hanna et al., 2020*, can serve as tools for parameterizing models (*Malagón et al., 2022*) or summarizing the impact of time to treatment across common cancers (*Gheorghe et al., 2021*), they may not capture information relevant to particular populations and outcomes. Three included meta-analyses (*Wang et al., 2016*; *Petrelli et al., 2016*; *Du et al., 2018*), which reported improved pCR rates among patients with advanced-stage rectal cancer who experienced time between NACRT and surgery of 6–8 weeks, demonstrated the biological benefit of lag time within the context of the intervention. Similarly, extended lag time to subsequent steps in care can be clinically appropriate with regard to patient rehabilitation post-treatment. The impact of lag time to diagnosis and/or treatment may also vary across different types of cancer with differing risk or rates of development within the same site (e.g., breast cancer). That is, just as prolonged lag time can be deleterious, which is often how it is connoted, it can also denote advantageous clinical characteristics. Our scoping review aimed to attend to the same need of summarizing lag time's impact on oncologic outcomes, however, with the intention of preserving the granularity of these associations across multiple cancer sites, with clearly mapped lag time interval start and endpoints.

Beyond potential changes in the true association between lag time and cancer-related outcomes attributable to period effects, such as that of the COVID-19 pandemic, we recognize that this true association may change over time with the advancement of treatment regimens and technology in cancer care. Just as the association between lag time to cancer care can change over time due to the advancement of other factors influencing patient outcomes (e.g., treatment modality), the same time-varying factors may have changed due to pandemic disruptions. Moreover, these alterations to additional factors could vary by country, health system, and wave of the pandemic. As such, direct comparison between risk estimates presented herein cannot be made with risk estimates of the association between lag time and patient outcomes during the pandemic when made publicly available. Instead, our tracing of defined lag times across relevant systematic reviews and/or meta-analyses can serve as a pre-pandemic reference when assessing the deviation in lag time duration attributable to the pandemic. Mapping the established relationships between lag time to care and oncologic outcomes when other elements of care were standard can help in constructing guidelines for flexible cancer care in the event of future public health emergencies. Most importantly, this scoping review could lay the groundwork for observational and meta-analytical research on lag time intervals' influence on oncologic outcomes across sites beyond the most common ones, such as breast, lung, and colorectal. Our detailed exploration into the methodological gaps in lag time literature can assist in identifying key statistics to report such as the distribution of lag time duration experienced by patients, stratified estimates by treatment regimen, and adjustment for time-varying factors that altered during the pandemic. As research is being conducted on the impact of the COVID-19 pandemic on the association between lag time and outcomes among patients with cancer, we believe that our findings can serve as guidance for the variables of interest in new studies.

## Conclusion

Through the aggregation of known associations between lag time and oncologic outcomes and exploration into gaps in lag time research, this scoping review can guide future studies and meta-analyses in the discipline. Our lag time interval timeline, or mapping of lag times on the cancer care continuum, emphasizes the granularity of exposure classification in cancer care. This timeline can act as a blueprint for future studies assessing lag time intervals and/or multiple cancer sites. With regard to the COVID-19 pandemic, our extensive characterization of the effect of lag time on oncologic outcomes could aid in gauging lag times in cancer care experienced during the pandemic.

## Acknowledgements

The present work was supported by the Canadian Institutes of Health Research (CIHR-COVID-19 Rapid Research Funding opportunity, VR5-172666 grant to Eduardo L Franco). Parker Tope, Eliya Farah, and Rami Ali each received an MSc stipend from the Gerald Bronfman Department of Oncology, McGill University.

## Additional information

### Competing interests

Parker Tope: received MSc. stipends from the Gerald Bronfman Department of Oncology, McGill University. Eliya Farah, Rami Ali: received an MSc. stipend from the Gerald Bronfman Department of Oncology, McGill University. Mariam El-Zein: reports a patent related to the discovery "DNA methylation markers for early detection of cervical cancer", registered at the Office of Innovation and Partnerships, McGill University, Montreal, Quebec, Canada. Wilson H Miller: reports grants to his institution from Merck, Canadian Institutes of Health Research, Cancer Research Society, Terry Fox Research Institute, Samuel Waxman Cancer Research Foundation, and CCSRI; consultancy for Merck, BMS, Roche, GSK, Novartis, Amgen, Mylan, EMD Serono, and Sanofi; honoraria from McGill University, JGH, BMS, Merck, Roche, GSK, Novartis, Amgen, Mylan EMD Serono, and Sanofi; payments to his institution for participation in a clinical trial within the past 2 years BMS, Novartis, GSK, Roche, AstraZeneca, Methylgene, MedImmune, Bayer, Amgen, Merck, Incyte, Pfizer, Sanofi, Array, MiMic, Ocellaris Pharma, Astellas, Alkermes, Exelixis, VelosBio, and Genetech. Eduardo L Franco: reports support for the present manuscript in the form of a grant to his institution is his name from the Canadian Institutes of Health Research and the Cancer Research Society; consultancy for Merck; a patent related to the discovery "DNA methylation markers for early detection of cervical cancer", registered at the Office of Innovation and Partnerships, McGill University, Montreal, Quebec, Canada; and financial interests with Elsevier and Elifesciences Ltd in the form of support fees to maintain the editorial office and work as Senior Editor, respectively.

### Funding

| Funder | Grant reference number | Author |
|---|---|---|
| Canadian Institutes of Health Research | VR5-172666 | Eduardo L Franco |
| McGill University | Gerald Bronfman Department of Oncology - MSc Stipend | Parker Tope Eliya Farah Rami Ali |

The funders had no role in study design, data collection, and interpretation, or the decision to submit the work for publication.

### Author contributions

Parker Tope, Conceptualization, Data curation, Formal analysis, Validation, Investigation, Visualization, Methodology, Writing – original draft, Writing – review and editing; Eliya Farah, Conceptualization, Data curation, Visualization, Methodology, Writing – review and editing; Rami Ali, Data curation, Methodology, Writing – review and editing; Mariam El-Zein, Conceptualization, Data curation, Supervision, Funding acquisition, Visualization, Methodology, Project administration, Writing – review and editing; Wilson H Miller, Eduardo L Franco, Conceptualization, Supervision, Funding acquisition, Project administration, Writing – review and editing

### Author ORCIDs

Parker Tope http://orcid.org/0000-0001-7903-1413
Mariam El-Zein http://orcid.org/0000-0002-5190-0370
Eduardo L Franco http://orcid.org/0000-0002-4409-8084

### Decision letter and Author response

Decision letter https://doi.org/10.7554/eLife.81354.sa1

Author response https://doi.org/10.7554/eLife.81354.sa2

## Additional files

### Supplementary files

• Supplementary file 1. Search strategy used to identify relevant systematic reviews and meta-analyses on the association between time to cancer diagnosis and treatment and outcomes of interest. The search was performed on 15 February 2021, limiting to publications from before the COVID-19 pandemic (1 January 2010–31 December 2019), with no restriction on publication language.

• Supplementary file 2. Characteristics of included systematic reviews on the association between time to cancer diagnosis and treatment and clinical outcomes.

• Supplementary file 3. Characteristics of and subgroup and/or sensitivity analysis reported by included meta-analyses on the association between time to cancer diagnosis and treatment and clinical outcomes. AC, adjuvant chemotherapy; CI, confidence interval; DFS, disease-free survival; HR, hazard ratio; LR, local recurrence; OR, odds ratio; OS, overall survival; pCR, pathological complete response; RCT, randomized controlled trial; RR, risk ratio; SMM, smoldering multiple myeloma. Significant pooled risk estimates are bolded. -- indicates that subgroup and/or sensitivity analyses were not conducted or were not available.

• MDAR checklist

### Data availability

This is a scoping review of peer-reviewed scientific literature. Data used came from scientific manuscripts which can be accessed online. All relevant information is included in the manuscript.

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
