## [Editor Report]

The scoping review has been constructed with novel time references for specific cancer treatment progressions. The extent of novel thought aggregating the literature makes an outstanding scientific contribution.

---

## [Decision Letter]

**Decision letter after peer review:**

Thank you for submitting your article "The impact of lag time to cancer diagnosis and treatment on clinical outcomes before the COVID-19 pandemic: a scoping review of systematic reviews and meta-analyses" for consideration by *eLife*. Your article has been reviewed by one peer reviewer, and the evaluation has been overseen by a Reviewing Editor and Diane Harper as the Senior Editor. The following individual involved in the review of your submission has agreed to reveal their identity: Philip Boonstra.

Recommendations for the authors:

– The use of the word 'contradictory' seems, at times, inappropriate. For example, in section T32 and ovarian cancer, it is stated that the two main meta-analyses arrived at contradictory findings because one found a statistically significant increase in mortality due to increases in lagtime and the other did not. It is, in my opinion, not appropriate to describe a non-statistically significant finding as contradicting a statistically significant finding. Both findings could very well be consistent under a null setting or a non-null setting.

– The connection to the Covid-19 pandemic is somewhat opaque. More specifically, although it stands to reason that the Covid-19 pandemic increased these various lag times (which the authors note on page 16, lines 486-487), it seems that the authors are also implying that the Covid-19 pandemic may have changed the *association* between lag-time and outcome. This is not stated explicitly, but the focus on characterizing relative risks would suggest that this is the case. If the former (namely, the pre-covid-19 distribution of lag times) is of interest, then really the most relevant statistic from the pre-Covid era is not the various associations between lag times and outcome but the distribution of lag times, which are only briefly mentioned and discussed in the manuscript. To serve as a benchmark for lag times, this manuscript would need to provide a carefully curated set of lag-time statistics from these various reviews and meta-analyses.

– A related question I have is how the relative risks from the pre-covid era are intended to be used as a benchmark. If the Covid-19 pandemic upset all aspects of patient care, then not only the conditional association between outcomes and lagtime will be modified but also (and more fundamentally) the baseline rates of outcomes. I recognize that such statistics are not readily available from meta-analyses or systematic reviews, and this leaves me wondering if the authors can more precisely characterize how the work as presently conducted will be able to help cancer patients on this side of the pandemic.

---

## [Author Response]

Recommendations for the authors:– The use of the word 'contradictory' seems, at times, inappropriate. For example, in section T32 and ovarian cancer, it is stated that the two main meta-analyses arrived at contradictory findings because one found a statistically significant increase in mortality due to increases in lagtime and the other did not. It is, in my opinion, not appropriate to describe a non-statistically significant finding as contradicting a statistically significant finding. Both findings could very well be consistent under a null setting or a non-null setting.

We agree with Dr. Boonstra’s concern regarding the language and particular word choice of ‘contradictory’. We revised the specific example he provided as is indicated below:

“The two meta-analyses on the association between time to ACT after surgery (T32) and risk of mortality among patients with ovarian cancer reported distinct findings; one found a significantly increased risk of mortality due to longer T32 (46), whereas the other did not find this relationship significant (45).”

We had also mentioned that findings from other systematic reviews were ‘contradictory’ in the manuscript’s Abstract. We thus amended the language (replacing “contradictory” with “different”) for appropriateness.

– The connection to the Covid-19 pandemic is somewhat opaque. More specifically, although it stands to reason that the Covid-19 pandemic increased these various lag times (which the authors note on page 16, lines 486-487), it seems that the authors are also implying that the Covid-19 pandemic may have changed the association between lag-time and outcome. This is not stated explicitly, but the focus on characterizing relative risks would suggest that this is the case.

We thank the reviewer for this nuanced suggestion as to the interpretation of our juxtaposition between pre-COVID-19 pandemic lag times and those experienced during the pandemic. Indeed, by specifically presenting our findings within the context of the COVID-19 pandemic, we aimed to summarize what and how past research on lag times in cancer care has been conducted across cancer sites as well as identify lag times in cancer care that would be most useful to investigate over the duration of the pandemic.

Specifically, our focus on characterization of relative risks, which were the statistics available from included systematic review and meta-analyses, was to synopsize the established risk of adverse oncologic outcomes attributable to lag time distinct treatment endpoints; we concentrated on the associations between time to treatment and risks of poor outcomes. We do not know if the association between time to treatment and cancer outcomes has changed since the start of the pandemic because other variables such as deviation from standard-of-care dosage and treatment may have changed in emergent periods of the pandemic. It is possible that the crisis induced changes to care standards of care could have influenced the magnitude or directionality of the associations between lag time and cancer-related outcomes during the pandemic period. However, we are confident that reporting relative estimates of risk, despite confounding changes, could provide a reference for the magnitude of change for a given lag time-oncologic outcome association attributable to the pandemic when it is eventually investigated.

Beyond potential changes in the true association between lag time and cancer-related outcomes attributable to period effects, such as that of the COVID-19 pandemic, we recognize that this true association may change over time with the advancement of treatment regimens and technology in cancer care. We acknowledge this limitation in our discussion and highlight the need for future research to stratify estimates by treatment type, as interventions varying in efficacy can influence the summary estimates. We further address concerns about our presentation of the relationship between our findings and the COVID-19 pandemic in Response #4.

If the former (namely, the pre-covid-19 distribution of lag times) is of interest, then really the most relevant statistic from the pre-Covid era is not the various associations between lag times and outcome but the distribution of lag times, which are only briefly mentioned and discussed in the manuscript. To serve as a benchmark for lag times, this manuscript would need to provide a carefully curated set of lag-time statistics from these various reviews and meta-analyses.

We are unsure as to whether Dr. Boonstra is referring to the “distribution of duration within specific lag time intervals” or “overall distribution of lag time intervals” experienced by patients with cancer over time.

Concerning the former potential interpretation of Dr. Boonstra’s comment, we agree that this would more appropriately distinguish between what lag time durations were standard prior to the pandemic and better characterize the variation in these standard lag time interval durations. However, the included systematic reviews and/or meta-analyses only provided the time range of the lag times experienced by patients with cancer based on the reporting in each individual study.

As for the latter interpretation of Dr. Boonstra’s comment, we were limited to mapping and presenting only lag time intervals for which their association with oncologic outcomes were systematically reviewed and/or meta-analyzed. We had restricted inclusion of literature to systematic reviews and/or meta-analyses as to select the highest level and quality of summarized evidence possible, however, we recognize the potential for differential selection and thus presentation of lag times which have been most researched. We endeavored to exhibit the distribution of these researched lag times in Figure 2, where we visualized the various lag time intervals included in our review across the cancer care continuum, as well as in Table 1, where we display at what frequency particular lag time intervals had been investigated across cancer sites.

To clarify this aim, we revised the manuscript to include the following:

“Table 1 highlights the distribution of and frequency at which unique lag time intervals were investigated over different cancer sites, and thus synopsizes the distribution of lag time intervals across systematic review and meta-analytical research.”

In order to attend to the limitations in reporting the distribution of lag time durations experienced by patients, we included the following revisions to the limitations section of our Discussion (red text designates changes):

“While we did not conduct quality assessment due to the nature of the included literature, we did describe characteristics of each review which could aid in assessing the validity and generalizability of each review’s findings. Moreover, differences in reporting across included systematic reviews and meta-analyses prevented our ability to provide (1) aggregated summary statistics of the associations between lag time and oncologic outcomes, and (2) the distribution of lag times experienced by patients included in each study within each meta-analysis. Our inability to provide these statistics stresses the need for transparent and comprehensive reporting in forthcoming research on lag times in cancer care.”

As for the second part of the Reviewer’s comment, two issues were raised. First, we wish to clarify that the purpose of the term ‘benchmark’ with regards to lag times was to emphasize the ‘clinically acceptable’ lag times experienced by patients with cancer prior to the pandemic in comparison to extended lag times during the COVID-19 pandemic. These ‘clinically acceptable’ lag time durations are presented in the ‘Lag time interval’ columns of Tables 3 and 4. We felt that presenting the range of lag times experienced by patient populations and the categorical cut-offs chosen by clinical investigators would capture what was standard and common prior to the pandemic. However, to Dr. Boonstra’s point, the term ‘benchmark’ could be misleading for readers as it connotes the ability for direct comparison between risk estimates reported in our scoping review and risk estimates reported during the pandemic. We concur. To avoid this possible misinterpretation, we replaced the term “benchmark” with “contextual reference”, as per the below modifications:

“Hence, the perspectives presented herein on the impact of lag time in cancer control and care not only aid in providing a contextual reference for pandemic-induced lag time compared to standard-of-care lag time experienced prior to the pandemic, but also inform ongoing research on these unprecedented lag times experienced by patients. The reported risk estimates therefore represent the estimated associations between lag time and cancer outcomes prior to the pandemic, which can serve as a standard-of-care reference of these associations regardless of other changes to cancer care due to the COVID-19 pandemic.”

Second, lag time statistics – such as the distribution of lag times and/or summary statistics aggregating the relative risks reported by included systematic reviews and meta-analyses through a meta-analysis of these meta-analyses – may provide further utility in encapsulating the relationship between lag time and cancer outcomes. However, pooling such summary statistics would have been inappropriate given the substantial heterogeneity in methodology across the included systematic reviews and meta-analyses and was beyond the scope of this scoping review’s aim. Moreover, aggregation of lag time-oncologic outcomes associations could have resulted in aberrant findings due to potential confounding. As such, we presented in as much detail as possible the statistical decisions made in each meta-analysis to provide the reader with enough context to determine the validity and utility of the various estimates of these associations.

– A related question I have is how the relative risks from the pre-covid era are intended to be used as a benchmark. If the Covid-19 pandemic upset all aspects of patient care, then not only the conditional association between outcomes and lagtime will be modified but also (and more fundamentally) the baseline rates of outcomes. I recognize that such statistics are not readily available from meta-analyses or systematic reviews, and this leaves me wondering if the authors can more precisely characterize how the work as presently conducted will be able to help cancer patients on this side of the pandemic.

As was stated previously in our Response #3, we agree with this concern. Time to care was one of many cancer care elements that were disrupted during the COVID-19 pandemic. Because lag time was our exposure of interest, we concentrated our efforts towards presenting what lag times were standard of care prior to the pandemic. Even though other factors like treatment regimens may have changed during the pandemic, clearly defining standard-of-care lag time durations and the deviation in lag times that occurred before pandemic disruptions could aid clinicians in determining how much deviation from lag times was ‘normal’ before the pandemic compared to during the pandemic. We hope that exhibiting these estimates would help in generating guidelines for future public health emergencies that would interrupt cancer care.

After taking into consideration Dr. Boonstra’s comments and suggestions, we added the following to the discussion of our scoping review (red text designates changes).

“Beyond potential changes in the true association between lag time and cancer-related outcomes attributable to period effects, such as that of the COVID-19 pandemic, we recognize that this true association may change over time with the advancement of treatment regimens and technology in cancer care. Just as the association between lag time to cancer care can change over time due to the advancement of other factors influencing patient outcomes (e.g., treatment modality), the same time-varying factors may have changed due to pandemic disruptions. Moreover, these alterations to additional factors could vary by country, health system, and wave of the pandemic. As such, direct comparison between risk estimates presented herein cannot be made with risk estimates of the association between lag time and patient outcomes during the pandemic when made publicly available. Instead, our tracing of defined lag times across relevant systematic reviews and/or meta-analyses can serve as a pre-pandemic reference when assessing the deviation in lag time duration attributable to the pandemic. Mapping the established relationships between lag time to care and oncologic outcomes when other elements of care were standard can help in constructing guidelines for flexible cancer care in the event of future public health emergencies. Most importantly, this scoping review could lay the groundwork for observational and meta-analytical research on lag time intervals’ influence on oncologic outcomes across sites beyond the most common ones, such as breast, lung, and colorectal. Our detailed exploration into the methodological gaps in lag time literature can assist in identifying key statistics to report such as the distribution of lag time duration experienced by patients, stratified estimates by treatment regimen, and adjustment for time-varying factors that altered during the pandemic. As research is being conducted on the impact of the COVID-19 pandemic on the association between lag time and outcomes among patients with cancer, we believe that our findings can serve as guidance for the variables of interest in new studies.”